# How Do Serum Zonulin Levels Change in Gestational Diabetes Mellitus, Pregnancy Cholestasis, and the Coexistence of Both Diseases?

**DOI:** 10.3390/ijerph182312555

**Published:** 2021-11-29

**Authors:** Huri Güvey, Samettin Çelik, Canan Soyer Çalışkan, Zehra Yılmaz, Merve Yılmaz, Özlem Erten, Andrea Tinelli

**Affiliations:** 1Department of Obstetrics and Gynecology, Private Kütahya Parkhayat Hospital, Kütahya 43020, Turkey; 2Department of Obstetrics and Gynecology, Samsun Training and Research Hospital, Samsun 55090, Turkey; drsamettincelik97@gmail.com (S.Ç.); canansoyer@hotmail.com (C.S.Ç.); 3Obstetrics and Gynecology, Private Office, Samsun 61420, Turkey; z.a.yilmaz@gmail.com; 4Department of Endocrinology and Metabolic Diseases, Gazi State Hospital, Samsun 55200, Turkey; drmerveyilmaz@gmail.com; 5Department of Obstetrics and Gynecology, Kütahya Health Sciences University, Kütahya 43020, Turkey; ozlem.erten@ksbu.edu.tr; 6Department of Obstetrics and Gynecology Veris delli Ponti Hospital, Scorrano, 73020 Lecce, Italy; andreatinelli@gmail.com; 7Department of Obstetrics and Gynecology, Division of Experimental Endoscopic Surgery, Imaging, Technology and Minimally Invasive Therapy, Vito Fazzi Hospital, 73100 Lecce, Italy; 8Phystech BioMed School, Faculty of Biological & Medical Physics, Moscow Institute of Physics and Technology, State University, 141701 Moscow, Russia

**Keywords:** intrahepatic cholestasis of pregnancy, gestational diabetes mellitus, zonulin, pregnancy complications, pregnancy outcomes

## Abstract

We investigated the question of how serum zonulin levels change in intrahepatic cholestasis of pregnancy (ICP) and gestational diabetes mellitus (GDM) and, in the case of the coexistence of ICP and GDM, evaluated the eventual increase in zonulin plasmatic levels. Participants were enrolled for the study between 25 February 2021 and 20 August 2021. The prospective case-control study included: group 1 of 95 pregnant women diagnosed with ICP; group 2 of 110 pregnant women diagnosed with GDM; group 3 of 16 women diagnosed with both GDM and ICP; group 4 of 136 healthy pregnant women as the control group. The groups were compared in terms of age, body mass index (BMI), gravidity, parity, gestational week of delivery, plasma zonulin levels, delivery type, birth weight, first- and fifth-minute APGAR scores, newborn intensive care unit (NICU) admission, and meconium staining of amniotic fluid parameters. The results suggested that the plasma zonulin levels of ICP (group 1), GDM (group 2), and GDM with ICP (group 3) patients were higher than those of the healthy pregnant women of group 4 (*p* < 0.001). Among the patient groups, the highest median plasma zonulin levels were found in group 3 (110.33 ng/mL). Zonulin levels were also associated with the severity of ICP and adverse pregnancy outcomes. High serum zonulin levels were related to GDM, ICP, and adverse perinatal outcomes. The coexistence of GDM and ICP led to higher serum zonulin concentrations.

## 1. Introduction

Intrahepatic cholestasis of pregnancy (ICP) is the most prevalent pregnancy-specific liver disease, characterized by itching, liver dysfunction, and/or elevated bile acid levels (≥10 µmol/L) in the third trimester of pregnancy. ICP signs, symptoms, and abnormal laboratory findings generally improve after labor, even if they may relapse during subsequent pregnancy or contraception assumption [1].

ICP can also lead to serious fetal compromises such as amniotic fluid with meconium, respiratory distress and preterm labor, and fetal death in association with maternal complications, such as itching, nausea, loss of appetite, or pain in the right upper quadrant [2]. Especially when the serum bile acid level is above 100 µmol/L, the incidence of adverse perinatal outcomes can significantly increase. ICP etiology has not been fully understood, but genetic, environmental, and hormonal factors can influence the development and exacerbation of this disease [3]. Gestational diabetes mellitus (GDM) is another metabolic disease of pregnancy characterized by glucose intolerance, resulting in maternal and fetal morbidity and mortality, as well as increasing the risk of diabetes mellitus during life [4]. Bile acids regulate cholesterol, glucose metabolism, and intestinal epithelial barrier characteristics via farnesoid X receptors (FXR) and shape intestinal microbiota [5,6]. Therefore, when women show bile acid dysregulation, they are susceptible to other metabolic disorders. Previous studies have demonstrated the coexistence of GDM and ICP [7,8]. Intestinal permeability is another essential factor regulating lipid and glucose metabolism, plus inflammation processes [9]. Intestinal barrier disruption and increased intestinal permeability can lead to metabolic and chronic inflammatory diseases, such as diabetes [10], cirrhosis [11], and non-alcoholic fatty liver disease [12].

Zonulin is the main protein regulating tight junctions between intestinal epithelial cells, mainly released from hepatocytes, enterocytes, adipose tissue, and immune cells [13]. The stimulation of a zonulin release leads to the dissociation of the zonula occludens-1 protein from tight junctions, resulting in increased intestinal permeability [14]. Disrupted intestinal barrier function is involved in the pathogenesis of these two metabolic diseases, GDM and ICP [15,16]. In this study, we investigated if serum zonulin levels change in ICP and GDM and, in the case of the coexistence of ICP and GDM, evaluated the eventual increase in zonulin levels.

## 2. Materials and Methods 

A prospective case-control study was conducted at the Health Sciences University Samsun Educational and Research Hospital. All pregnant women with ICP and GDM diagnoses were suitable for the study and were screened at the obstetrics outpatient clinic of the Health Sciences University Samsun Educational and Research Hospital for enrollment in the study between 25 February 2021 and 20 August 2021. They were age-matched between 18 and 45 years old and lived in the same geographical area. All participants signed a written informed consent form at enrollment. The study protocol was performed according to the principles of the Declaration of Helsinki and was approved by the local ethics committee of the Health Sciences University Samsun Educational and Research Hospital (approval number OMÜ KAEK 2021/96).

The gestational age was estimated based on the first day of the last menstruation and/or the first trimester ultrasound. Body mass index (BMI) was calculated as weight (in kilograms) divided by the square of height (in meters). All screened women received a physical examination and obstetric ultrasound, along with a complete blood count, routine biochemistry tests, 75 g oral glucose tolerance test (OGTT), hepatic tests, and urinalysis.

The exclusion criteria were the following: women with any infection, allergic diseases, chronic diseases (including types 1 and 2 diabetes mellitus), itching due to other etiologies, multiple pregnancies, having a fetus with malformation or aneuploidy, in active labor, or a history of medication use. The diagnosis of ICP was made based on the following criteria: (1) pruritus (late second or third trimester without other etiologies), (2) elevated serum bile acids (≥10 µmol/L), and (3) a two-fold or higher elevation in serum transaminases [17]. The patients whose total bile acid level was 10–100 μmol/L were classified as those with mild cholestasis, and those with ≥100 μmol/L were classified as severe cholestasis [3]. Blood was collected to determine the 8 h fasting total serum bile acid in patients who presented to the hospital with pruritus without primary skin lesions. The blood samples from which the maternal fasting total serum bile acids were analyzed were sent to the Düzen Laboratory in Ankara as a reference center. Other laboratory tests were performed in the biochemistry department of this hospital. Moreover, all pregnant women with ICP underwent hepatic ultrasound evaluation to discriminate against other hepatic disorders. Ursodeoxycholic acid (250 mg three times a day) was given to the patients who were diagnosed with ICP [18]. 

At 24–28 weeks of pregnancy, screening for GDM was performed on all participants using the 75 g OGTT, and the diagnostic criteria were based on the American Diabetes Association guidelines, including the following: fasting blood glucose level above 92 mg/dL, 75 g OGTT 1 h blood glucose level of ≥180 mg/dL, and 2 h blood glucose level above 153 mg/dL [19]. Appropriate treatment was provided to pregnant women diagnosed with GDM. Blood samples were obtained in the third trimester from all participants through venipuncture and processed within 1 h after withdrawal by centrifugation at 5000 rpm for 15 min. All serum samples were stored at −80 °C until the day of analysis.

The concentrations of human zonulin in sera were measured using commercially available enzyme-linked immunosorbent assay (ELISA) kits (Sun-Red Bio Company, Cat No. 201-12-5578, Shanghai, China). This kit uses a double-antibody sandwich ELISA to assay the level of human zonulin. The enzymatic reactions were quantified using an automatic microplate photometer. The microtiter plate provided in this kit was pre-coated with a monoclonal antibody specific to human zonulin. Standards or samples were then added to the appropriate microtiter plate wells with a biotin-conjugated antibody preparation specific for human zonulin. Streptavidin–horseradish peroxidase was added to each microplate well and incubated to form an immune complex. After chromogen solutions were added, only the wells that contained human zonulin, biotin-conjugated antibody, and enzyme-conjugated avidin exhibited a change in color. The enzyme-substrate reaction was terminated through the addition of a sulfuric acid solution. The color change was measured spectrophotometrically at a wavelength of 450 nm. The concentration of human zonulin in the samples was determined by comparing the optic density of the samples with the standard curve. The human zonulin levels were expressed as ng/mL. The sensitivity of this assay was 0.223 ng/mL, and the assay range was 0.25–70 ng/mL. All assays were conducted according to the manufacturer’s instructions. The samples, which showed high concentrations, were diluted and measured in duplicate. Finally, 342 participants were enrolled in the study and divided into four groups: 

Group 1: 95 pregnant women diagnosed with ICP;Group 2: 110 pregnant women diagnosed with GDM;Group 3: 16 women diagnosed with both GDM and ICP;Group 4: 136 healthy pregnant women as the control group.

Patients who were diagnosed with GDM and later complicated with ICP during follow-up were included in Group 3. For the patients with GDM and/or ICP, follow-ups were performed periodically until delivery by fetal ultrasound (US) scan (including Doppler US and biophysical profile) and laboratory analysis. If patients had worsening clinical presentations or laboratory results and/or any unfavorable evidence of fetal wellbeing, the delivery timing was managed. Pregnant women whose gestational week of delivery was below 37 were administered with two doses of betamethasone. Infants were categorized into three groups according to birth weight: (1) <2500 g, (2) 2500–3999 g, and (3) ≥4000 g. The endpoint of the study was delivery by the participants.

The groups were compared in terms of age, BMI, gravidity, parity, and gestational week of delivery, plasma zonulin levels, delivery type (cesarean section (CS) or vaginal delivery), parameters of mothers and birth weight, first- and fifth-minute APGAR scores, newborn intensive care unit (NICU) admission, and meconium staining of amniotic fluid parameters of newborns.

An experimental power analysis was carried out, taking into account the findings of the present study and using a large effect-size value (d = 0.5). The calculated effect size was determined as n = 86 units for each group, and the power value obtained from working with an error of proportion of 0.05 was 90%. Data analysis was performed using SPSS 25 (Statistical Package for Social Sciences Statistics for Windows, version 25.0. Armonk, NY, USA: IBM Corp.). Descriptive statistics were presented as the median (minimum-maximum) for numerical variables and as the number of observations and percentile for nominal variables. The variables were tested for normality using the Kolmogorov–Smirnov and Shapiro–Wilk tests. An independent-samples *t*-test was used to determine whether there was a statistically significant difference between the groups in terms of the numerical variables distributed normally. The Mann–Whitney U test and Kruskal–Wallis test were used to evaluate whether there was a statistically significant difference between the groups in terms of non-normally distributed numerical variables. The results for *p* < 0.05 were considered statistically significant. The Bonferroni-corrected Mann–Whitney U test was used to determine the significance originating from the variables found to be significant using the Kruskal–Wallis test, and the variables with *p* < 0.0083 were considered significant. The nominal variables were evaluated using the chi-squared test, Fisher’s exact test, and the Fisher–Freeman–Halton exact test. The associations between the variables were evaluated using Pearson’s correlation analysis. Logistic regression analysis was performed to calculate the odds ratios (OR) with a 95% confidence interval.

## 3. Results

Among the 341 women enrolled at the beginning of the study, three patients were excluded from the GDM group (one had hypothyroidism and two had chronic hypertension), and 16 in the GDM group were later diagnosed with ICP. Furthermore, seven pregnant women were excluded from the ICP group (two had chronic hepatitis B, one had epilepsy, one had psoriasis, and two had hypothyroidism). Finally, 331 pregnant women were included in the study (88 with ICP, 91 with GDM but without ICP, 16 GDM with ICP, and 136 patients as controls) (Figure 1).

No statistically significant difference in age (*p* = 0.127), gravidity (*p* = 0.724), and parity (*p* = 0.710) of the participants was detected between groups. Table 1 shows the baseline characteristics, plasma zonulin levels, and perinatal outcomes of the groups.

Pairwise comparisons were performed to understand from which group the significance originated in the variables found to be significant between the groups (Table 2). 

Fetuses weighing 4000 g and above were mostly seen in the GDM without ICP group (36.3%), whereas fetuses weighing lower than 2500 g were mostly found in the ICP group (42.1%) (Figure 2 and Table 3).

In the comparison between patients with mild and severe ICP, no significant difference was found in terms of age (*p* = 0.169), BMI (*p* = 0.057), gravidity (*p* = 0.166), parity (*p* = 0.229), fifth-minute APGAR score (*p* = 0.404), delivery type (*p* = 0.408), or gestational week of delivery (*p* < 0.001). Conversely, the first-minute APGAR score (*p* = 0.032) was lower in patients with severe ICP and SGA rate (*p* < 0.001), whereas plasma zonulin levels (*p* < 0.001), NICU admission rate (*p* < 0.001), and meconium staining of amniotic fluid (*p* < 0.001) were lower in patients with mild ICP (Table 4).

When we further categorized all participants according to BMI levels as <25, 25–29, and ≥30; zonulin concentrations showed significant difference between groups (Table 5). Although the zonulin level of pregnant women with BMI < 25 showed no significant difference between groups (*p* = 0.601), that of pregnant women with BMI = 25–29 (*p* = 0.015) and ≥30 presented significant differences (*p* < 0.001) in favor of obesity.

After considering all groups, a significant negative correlation was found between the first- (r= −0.304; *p* < 0.001) and fifth-minute APGAR (r= −0.286; *p* < 0.001) scores and plasma zonulin levels. Moreover, a significant positive correlation was observed between zonulin levels and NICU admission rate (*p* < 0.001; OR:1.022; CI:1.014–1.031) and the risk of amniotic fluid staining with meconium (*p* < 0.001; OR:1.029; CI:1.020–1.039).

## 4. Discussion

The results suggested that the plasma zonulin levels of ICP (group 1), GDM (group 2), and GDM with ICP (group 3) patients were higher than those of healthy pregnant women (group 4). Among the patient groups, the highest plasma zonulin levels were found in group 3. Zonulin levels were associated with the severity of ICP and adverse pregnancy outcomes. Previous studies have demonstrated the relationship between GDM and zonulin. In a prospective study on 88 pregnant women, the serum zonulin level was associated with GDM. Sensitivity was 88% and specificity was 47% for 43.3 ng/mL cut-off value. The authors suggested that zonulin is a good predictor of GDM [20].

Similarly, a prospective study with a larger sample size investigated the plasma zonulin levels of 314 pregnant women and applied 50 g OGTT to the participants at 24–28 gestational weeks. Pregnant women diagnosed with GDM had significantly higher plasma zonulin levels than healthy participants. The cut-off value for zonulin was 47.5 ng/mL, with 80.95% sensitivity and 80.41 specificity [16].

A cross-sectional study of 100 overweight pregnant women, conducted by Mokkala et al. [21], showed that the serum zonulin concentration was positively correlated with inflammatory markers, insulin, and insulin resistance. A prospective observational study including 85 pregnant women diagnosed with GDM and 90 pregnant women without GDM, showed that the plasma zonulin levels of patients diagnosed with GDM were significantly higher than those of the control group. These results were consistent with the abovementioned research and this investigation [22].

In the current study, the plasma zonulin levels in pregnant women with ICP were also investigated, and the results showed significantly higher values than those of healthy participants. Reyes et al. [15] showed that “leaky gut” could contribute to the pathogenesis of ICP. The authors evaluated the urine lactulose/mannitol ratio (L/M) to examine gastrointestinal permeability. The L/M ratio and interleukin-6 levels of pregnant women with ICP were found to be significantly higher than those of the control group. The authors concluded that a “leaky gut” could lead to an increase in the absorption of endotoxins and bile salts, promoting inflammation in the liver and resulting in ICP.

Deniz et al. were the first to reveal the association between zonulin and ICP in the literature [23]. They evaluated the plasma zonulin levels and perinatal outcomes of 44 pregnant women with ICP and 44 controls. They found that the plasma zonulin levels of ICP patients were significantly higher than those of the control group and that severe ICP patients had significantly higher zonulin levels than mild ICP patients. The authors showed a correlation between zonulin levels and adverse pregnancy outcomes, consistent with study results, in which the highest zonulin levels were found in the GDM with ICP group. This result can be explained by the pathogenesis of ICP and GDM, which are based on complex mechanisms.

The intestinal barrier is disrupted in both ICP and GDM [15,22]. Therefore, antigens and other macromolecules can pass through the intestinal epithelium into the body, mainly using paracellular pathways, leading to systemic or local inflammation. Tight junctions between intestinal epithelial cells regulate paracellular transport [24]. The dissociation of tight junctions by zonulin through epidermal growth factor receptor activation increases intestinal permeability [25]. Systemic low-grade inflammation triggered by zonulin leads not only to chronic inflammatory diseases, such as multiple sclerosis, celiac disease, and ankylosis spondylitis, but also to GDM and ICP [23,26,27].

Supported by these findings, other studies have indicated an association between GDM and ICP. In a retrospective study, Martinaeu et al. [7] evaluated the data of 57,724 pregnant women. They concluded that the incidence of GDM was higher in women who were susceptible to ICP. Similarly, according to the results of a meta-analysis, pregnancies complicated by ICP were more likely to develop GDM (OR = 2.19) [8]. In contrast with the abovementioned studies, our study was prospective, and the patients were initially diagnosed with GDM and later with ICP.

In the light of our analysis and the relevant literature, zonulin seems to be one of the shared points in the pathogenesis of both GDM and ICP. Furthermore, the zonulin level was found to be related with BMI, which is consistent with the current knowledge [28]. It was detected in our results that the highest BMI level belonged to pregnant women having both GDM and ICP, rather than those having GDM alone. This result is consistent with the current literature, which indicates obesity as a risk factor for GDM [29]. It was inferred from the results that obesity might trigger the main metabolic diseases of pregnancy by inducing different pathophysiological mechanisms. This implication suggests the importance of weight control in pregnancy.

According to the outcomes of the current study, pregnant women with ICP had higher rates of meconium-stained amniotic fluid, lower gestational weeks of delivery and due to this, lower birth weight APGAR scores, in agreement with the existing literature, and higher Caesarean delivery rates and similar NICU admission rates with those of the control group, in contrast with other study results [30,31,32]. However, similarly to other investigation results, patients with GDM had higher NICU admission rates and lower APGAR scores, but similar gestational weeks of delivery to the control group [33,34]. These differences between studies might originate from the selection of different study populations and diagnostic criteria of GDM.

Infants with birth weights lower than 2500 g were most often seen in group 3, due to their lower gestational age at delivery. Group 2 had the highest number of infants heavier than 4000 g due to macrosomia complications of GDM.

This investigation has some limitations, as it was conducted in a single tertiary center and was not adjusted for the participants’ dietary factors. The study evaluated only the serum zonulin concentrations, but not the fecal zonulin levels. Most of published studies examined only the zonulin concentrations [20,22,23] in plasmatic samples; thus, the authors compared results with these studies. Furthermore, because BMI is not an adjustable and changeable parameter, it constitutes a confounding factor. Due to the rarity of pregnant women having ICP and GDM and ICP, the study population had to consist of participants with heterogeneous BMI levels. The other limitation is the lack of a further classification of pregnant women with GDM according to disease severity. Apart from these limitations, zonulin is identified as precursor of haptıglobulin-2 (prehaptoglobulin-2) and there are different genetic variants of zonulin and haptoglobin that have been identified [13]. Due to this, commercially available immunodiagnostic ELISA kits such as the one we used in this study, have the potential to detect not only pure prehaptoglobulin-2, but also a variety of proteins that are structurally and functionally relevant to zonulin [35]. This study should be interpreted taking into account this limitation.

The strengths of this study are that the sample size was larger than most similar studies, which provides more generalizability than others. Moreover, to our knowledge, this study is the first to examine the serum zonulin levels of pregnant women diagnosed with GDM with ICP and to compare them separately with the GDM without ICP, ICP, and healthy control groups.

## 5. Conclusions

High serum zonulin levels were related to GDM, ICP, and adverse perinatal outcomes. The coexistence of GDM and ICP led to higher serum zonulin concentrations. Zonulin could exist in the pathogenesis of GDM and ICP separately and together. Furthermore, it can be speculated that new treatments targeting zonulin might improve the severity and perinatal outcomes of both diseases. Further longitudinal prospective studies comparing serum and fecal zonulin levels in early pregnancy and in the diagnostic period of GDM and ICP could support the relevant literature.

## Figures and Tables

**Figure 1 ijerph-18-12555-f001:**
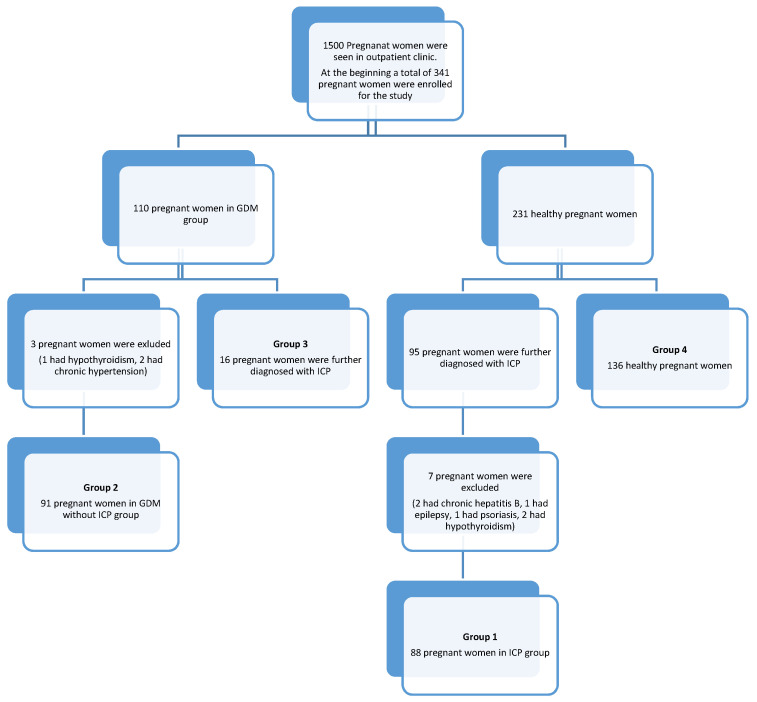
Flowchart of the study.

**Figure 2 ijerph-18-12555-f002:**
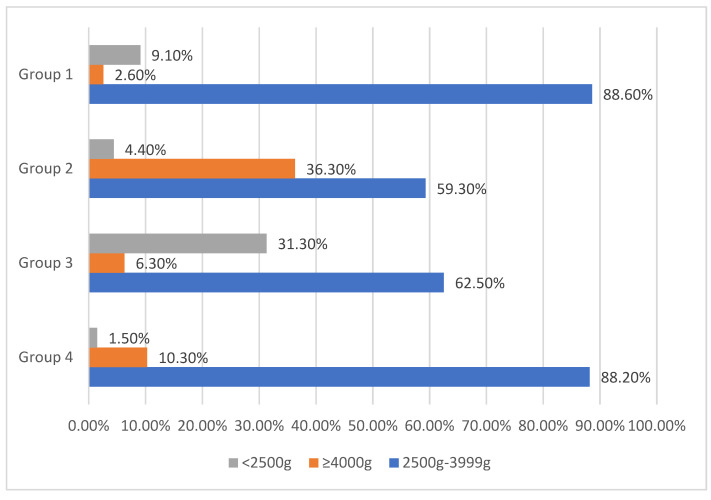
Comparison of groups regarding birth weight into three categories.

**Table 1 ijerph-18-12555-t001:** Baseline characteristics, plasma zonulin levels, and perinatal outcomes of groups.

Variables	Group 1 (n = 88)	Group 2 (n = 91)	Group 3 (n = 16)	Group 4 (n = 136)	*p* Value
Age (years) median (min–max)	26.5 (18–41)	27 (20–42)	32 (18–39)	26 (18–45)	0.127 ^a^
BMI (kg/m^2^) median (min–max)	27.71 (20.12–39.12)	30.12 (22.12–38.12)	35.39 (27.88–39.12)	26.38 (20.22–42.12)	**<0.001 ^a^**
Gravidity median (min–max)	1 (1–6)	1 (1–5)	1 (1–6)	2 (1–8)	0.724 ^a^
Parity median (min–max)	0 (0–4)	0 (0–4)	0 (0–3)	1 (0–5)	0.710 ^a^
Gestational week of delivery median (min–max)	37 (34–40)	39 (28–40)	36 (34–38)	39 (30–42)	**<0.001 ^a^**
1st minute APGAR score median (min–max)	9 (5–9)	8 (5–9)	8 (5–9)	9 (5–9)	**<0.001 ^a^**
5th minute APGAR score median (min–max)	9 (8–10)	10 (7–10)	9 (8–10)	10 (8–10)	**<0.001 ^a^**
Zonulin (ng/mL) median (min–max)	12.11 (1.77–149.12)	31,03 (0.77–103.11)	110.33 (66.12–188.9)	4.77 (0.12–101.11)	**<0.001 ^a^**
Delivery type (CS/VD) n (%)	37/51 (42%/58%)	31/60 (34.1%/65.9%)	3/13 (18.8%/81.3%)	27/109 (19.9%/80.1%)	**0.002 ^b^**
NICU admission n (%)	7 (8%)	19 (20.9%)	3 (18.8%)	8 (5.9%)	**0.003 ^b^**
Meconium staining n (%)	8 (9.1%)	10 (11%)	8 (50%)	9 (6.6%)	**<0.001 ^b^**

The levels of categories are presented as median (min–max) for numerical variables and as the number of observations and percentile for nominal variables. Values in bold represent statistically significant outcomes. Abbreviations: NICU: newborn intensive care unit CS: Caesarean section VD: vaginal delivery BMI: body mass index. ^a^ Kruskal–Wallis test. ^b^ Pearson chi-squared test.

**Table 2 ijerph-18-12555-t002:** *p* values of pairwise comparisons of groups that provided significant results.

Variables	Group 1–Group 2	Group 1–Group 3	Group 1–Group 4	Group 2–Group 3	Group 2–Group 4	Group 3–Group 4
BMI (kg/m^2^)	**<0.001**	**<0.001**	0.520	**<0.001**	**<0.001**	**<0.001**
Gestational week of delivery	**<0.001**	**0.001**	**<0.001**	**<0.001**	0.933	**<0.001**
1st minute APGAR score	**<0.001**	**0.025**	**<0.001**	0.772	**<0.001**	**<0.001**
5th minute APGAR score	**<0.001**	0.209	**<0.001**	**<0.001**	0.973	**<0.001**
Zonulin	**<0.001**	**<0.001**	**<0.001**	**<0.001**	**<0.001**	**<0.001**
Delivery type (CS)	0.271	0.138	**<0.001**	0.356	0.024	1
NICU admission	0.025	0.181	0.740	1	**0.001**	0.094
Meconium staining	0.862	**<0.001**	0.671	**0.001**	0.357	**<0.001**

Values in bold represent statistically significant outcomes. Abbreviations: NICU: newborn intensive care unit CS: Caesarean section BMI: body mass index. *p* values were calculated with the Bonferroni corrected Mann–Whitney U test. Fetuses weighing 2500–3999 g were the most common in all groups (*p* < 0.001).

**Table 3 ijerph-18-12555-t003:** Comparison of groups regarding birth weight into three categories.

Groups	2500 g–3999 g (n = 262)	≥4000 g (n = 50)	<2500 g (n = 19)	*p*
Group 1 (n = 88)	78 (88.6%)	2 (2.3%)	8 (9.1%)	**<0.001**
Group 2 (n = 91)	54 (59.3%)	33 (36.3%)	4 (4.4%)	
Group 3 (n = 16)	10 (62.5%)	1 (6.3%)	5 (31.3%)	
Group 4 (n = 136)	120 (88.2%)	14 (10.3%)	2 (1.5%)	

*p* values were calculated with the Pearson chi-squared test. Values in bold represent statistically significant outcomes.

**Table 4 ijerph-18-12555-t004:** Comparison of baseline characteristics, plasma zonulin levels, and perinatal outcomes according to severity of intrahepatic cholestasis of pregnancy.

Variables	Mild ICP (n = 80)	Severe ICP (n = 24)	*p*
Age (years) median (min–max)	28 (18–41)	29 (19–41)	0.169 ^a^
BMI (kg/m^2^) median (min–max)	27.77 (20.12–39.12)	29.11 (23.33–39.12)	0.057 ^b^
Gravidity median (min–max)	1 (1–6)	1,5 (1–6)	0.166 ^b^
Parity median (min–max)	0 (0–3)	0,5 (0–4)	0.229 ^b^
Gestational week of delivery median (min–max)	38 (35–40)	36 (34–37)	**<0.001 ^b^**
1. minute APGAR score median (min–max)	9 (6–9)	8 (5–9)	**0.032 ^b^**
5. minute APGAR score median (min–max)	9 (8–10)	9 (8–10)	0.404 ^b^
Zonulin (ng/mL)	11.84 (1.77–116.12)	83.27 (5.81–188.9)	**<0.001 ^b^**
Delivery type (CS/VD) n (%)	33/47 (41.2%/58.8%)	7/17 (29.2%/70.8%)	0.408 ^c^
NICU admission n (%)	0 (0%)	10 (41.7%)	**<0.001 ^c^**
Meconium staining n (%)	2 (2.5%)	14 (58.3%)	**<0.001 ^c^**
Birth weight (2500 g–3999 g /≥4000 g/S < 2500 g) n (%)	75/3/2 (93.8%/3.8%/2.5%)	13/0/11 (54.2%/0%/45.8%)	**<0.001 ^c^**

The levels of categories are presented as the median (min–max) for numerical variables and as the number of observations and percentile for nominal variables. Values in bold represent statistically significant outcomes. Abbreviations: NICU: newborn intensive care unit CS: Caesarean section VD: vaginal delivery BMI: body mass index. ^a^ independent samples *t* test. ^b^ Kruskal–Wallis test. ^c^ Pearson chi-squared test.

**Table 5 ijerph-18-12555-t005:** Comparison of zonulin concentrations according to BMI levels.

BMI Levels	Group 1	Group 2	Group 3	Group 4	*p* Value
<25	12.24 (1.77–99.12)	10.50 (6.00–15)	No data	4.9 (10.12–23.88)	0.601
25–29	11.13 (2.33–149.12)	18.5 (2.33–101.20)	108.12 (66.12–150,12)	6.7 (0.16–150.12)	**0.015**
≥30	15.09 (2.88–116.12)	44.66 (0.77–103.11)	110.33 (77.32–188.90)	7.5 (0.26–101.11)	**<0.001**

The levels of categories are presented as the median (min–max) for numerical variables and as the number of observations and percentile for nominal variables. The Kruskal–Wallis test was applied. Values in bold represent statistically significant outcomes. Abbreviations: BMI: body mass index.

## Data Availability

The data presented in this study are available on request from the corresponding author. The data ate not publicly available due to privacy restrictions.

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
