# Peer review of "How Do Serum Zonulin Levels Change in Gestational Diabetes Mellitus, Pregnancy Cholestasis, and the Coexistence of Both Diseases?"

_ijerph, 2021, doi:10.3390/ijerph182312555_

Round 1
Reviewer 1 Report
Dear Editor, Hi Thank you for the opportunity to review the manuscript.
I really liked the manuscript, as it deals with a very relevant subject for clinical practice.
Initially, I suggest that the authors use the STROBE Checklist instrument - Observational Studies (Case-Control Studies). This is essential for the study report to be clear and concise.
The liver is known to play a key role in the regulation of glucose homeostasis, so that diseases
liver can cause diabetes mellitus. GDM is a factor with unfavorable outcomes for mothers and newborns. How did the authors deal with this fact? It is a strong limitation, as unfavorable neonatal outcomes can occur due to GDM. There are some cases where GDM is not controlled and insulin is required, which makes the scenario even more complicated. In the present study, there is no mention of glycemic maintenance in pregnant women. I suggest the authors insert it in the discussion as a limitation.
In summary: Please change the tense.
Where it says "Authors investigated" replace it with "has been investigated".
Please insert the collection period in the summary.
in the method, when talking about the ethical factor: Please mention that the patients signed the term when they agreed to participate in the study.
At the beginning of your flowchart, please insert the number of patients seen in the service, as it starts with "341 pregnant women were enrolled for the study" but it is important to contextualize the sample universe of the service where the study was carried out.
In the flowchart, please insert a caption under the figure, as it facilitates the reader's understanding.
In the flowchart also insert groups 1,2,3 and 4. There is no mention in the figure about the formation of the study groups.
I congratulate the authors for the manuscript.
Success in publication.
Author Response
I really liked the manuscript, as it deals with a very relevant subject for clinical practice.
Initially, I suggest that the authors use the STROBE Checklist instrument - Observational Studies (Case-Control Studies). This is essential for the study report to be clear and concise.
Answers: Thank you for your valuable suggestion. We attached the Stobe Checklist of the manuscript.
The liver is known to play a key role in the regulation of glucose homeostasis, so that diseases liver can cause diabetes mellitus. GDM is a factor with unfavorable outcomes for mothers and newborns. How did the authors deal with this fact? It is a strong limitation, as unfavorable neonatal outcomes can occur due to GDM. There are some cases where GDM is not controlled and insulin is required, which makes the scenario even more complicated. In the present study, there is no mention of glycemic maintenance in pregnant women. I suggest the authors insert it in the discussion as a limitation.
Answers: Thank you for your valuable opinion that gives us the opportunity to explain this point in the article. As you mentioned, we didn’t classified the gestational diabetes patients according to disease severity. We can excluded only pregnant women with known or new diagnosed liver disease. We added this point as limitation to discussion section.
In summary: Please change the tense.
Where it says "Authors investigated" replace it with "has been investigated".
Please insert the collection period in the summary.
in the method, when talking about the ethical factor: Please mention that the patients signed the term when they agreed to participate in the study.
At the beginning of your flowchart, please insert the number of patients seen in the service, as it starts with "341 pregnant women were enrolled for the study" but it is important to contextualize the sample universe of the service where the study was carried out.
In the flowchart, please insert a caption under the figure, as it facilitates the reader's understanding.
In the flowchart also insert groups 1,2,3 and 4. There is no mention in the figure about the formation of the study groups.
I congratulate the authors for the manuscript.
Success in publication.
Answers: Thank you for your valuable suggestions and kind commentaries. We made all necessary cahanges on the manuscript and marked-up with yellow colour.

Reviewer 2 Report
Authors of the presented study assessed the concentration of zonulin in the serum of precisely described pregnant women with GDM (group no 1), ICP (group no 2) and both of these disorders (group no 3). They figured out that zonulin was mostly expressed in the serum of the last group. Findings of this survey do not belong to novelties; similar dependencies between zonulin, GDM and ICP were already described in the literature. Nevertheless, in comparison to previous surveys in this field, zonulin was measured in the relatively large group of pregnant patients and the presented combination of accompanying disorders (GDM, ICP or both of them) appears to be a quite novel idea. These last points constitute strong issues of the manuscript. After taking everything into consideration, it seems to me that the presented article is worth publishing; also because of the worldwide commonness and significance of GDM and ICP. To improve a scientific value of the manuscript, I would like authors to speculate about a potential role of the zonulin in the models of treatment of GDM and ICP.
Author Response
Authors of the presented study assessed the concentration of zonulin in the serum of precisely described pregnant women with GDM (group no 1), ICP (group no 2) and both of these disorders (group no 3). They figured out that zonulin was mostly expressed in the serum of the last group. Findings of this survey do not belong to novelties; similar dependencies between zonulin, GDM and ICP were already described in the literature. Nevertheless, in comparison to previous surveys in this field, zonulin was measured in the relatively large group of pregnant patients and the presented combination of accompanying disorders (GDM, ICP or both of them) appears to be a quite novel idea. These last points constitute strong issues of the manuscript. After taking everything into consideration, it seems to me that the presented article is worth publishing; also because of the worldwide commonness and significance of GDM and ICP. To improve a scientific value of the manuscript, I would like authors to speculate about a potential role of the zonulin in the models of treatment of GDM and ICP.
Answers: Thank you for your valuable opinion and contribution. We added a comment about treatment in conclusion section and marked-up with yellow colour. “Also it can be speculated that new treatments targeting zonulin might improve the severity and perinatal outcomes of both diseases”.

Reviewer 3 Report
The authors investigated the relationship between serum zonulin levels and two pregnancy-specific metabolic diseases, intrahepatic cholestasis of pregnancy (ICP) and gestational diabetes mellitus (GDM), through a prospective case-control study. A total of 342 participants were enrolled in four groups: 95 ICP, 110 GDM, 16 ICP and GDM, and 136 healthy control. Participants were age-matched, came from the same geographic area, and were screened for exclusion criteria like chronic diseases. Blood samples were obtained from all participants in the third trimester, and human zonulin concentrations in sera were measured using Sun-Red Bio Company ELISA kits. After delivery, birth weight of the newborns and other data were collected to compare parameters. Then, statistical analyses were performed to test for differences between the groups.
- Strengths:
- The study had a large sample size and collected information on a number of parameters that could affect patient outcomes.
- The authors provided extensive statistical data in tables containing group baseline characteristics, pairwise comparisons, and the significance of any differences.
- The study design was clearly explained. The authors explained the unique aspects of this study compared with others in the discussion section, like including a group for patients with both GDM and ICP.
- The authors acknowledge some limitations of the study, like not controlling for dietary factors or measuring fecal zonulin levels.
- Weaknesses:
Major
- The authors mention some limitations of the study but do not acknowledge the potentially confounding statistically significant difference in BMI between the groups, a factor that could affect “leaky gut” and zonulin levels regardless of ICP or GDM diagnosis. The group with the highest zonulin levels also has the highest BMIs.
- While there are many statistically significant results in the provided tables, the authors only really discuss the zonulin levels.It would be interesting to see their analysis of the other characteristics that differed between the groups.
- The discussion section was largely a summarization of other studies on this topic. While it may be relevant that the results were consistent with other studies, I would have expected a deeper analysis of the authors’ own results and their meaning in the context of ICP and GDM.
- The chosen ELISA Kit to measure Zonulin should be explained since measuring blood Zonulin is very difficult and critical too. Several Publications report on the disadvantages of this ELISA Kits.
Minor
- Line 3: “Coexistence”
- Line 8, 20: email font size larger than the rest
- Line 24: “intrahepatic” (lowercase i)
- Line 44: “liver disease, characterized by…”
- Line 47: I’m not sure what the authors mean by “contraception assumption”. Do they mean conception?
- Line 48: “ICP” instead of “The ICP”
- Line 50: “such as itching, nausea, loss of appetite, or pain in the…”
- Line 54: “Gestational diabetes mellitus” instead of “The gestational diabetes mellitus”
- Lines 60-61: The sentence feels incomplete: “In fact, previous studies have demonstrated the coexistence of GDM and ICP”. Even this study looks at people who have both GDM and ICP, proving that both can exist in the same person. Was there something more to this? Like a connection to the previous sentence about bile acids?
- Lines 75, 102, 230, 235, 236, 241, 247, 285: Replace “pregnants” with “participants”, “patients”, or “pregnant women”.
- Line 91: “including types 1” (missing a space)
- Line 94: “1) pruritus” (missing a space)
- Lines 104, 110: Replace “applied” with “given” or “provided”.
- Line 109: “mg/dL,” instead of “mg/dL.”
- Line 127: “zonulin in the samples” instead of “zonulin the samples”
- Line 138: “in Group 3” instead of “in the group 3”
- Line 138-139: “follow-ups were performed” instead of “follow-up was performed”
- Line 144: Consider listing weight groups in ascending order, with the first group the lowest weight (<2,500g) and the third group the highest (>4,000g).
- Line 192/Table 2: Consider ordering the rows left to right from lowest numerical comparison to highest. (Group 1-Group 2, then Group 1-Group 3, then Group 1-Group 4, and so on)
- Line 198/Figure 2: Consider listing the groups from top to bottom in ascending order (1,2,3,4) for consistency with other tables.
- Line 207: “(p=0.404)” instead of “(0.404)”
- Line 235: “gestational weeks” instead of “gestational week”
- Line 259: “study results” instead of study’ results”
- Line 280: “participants’ dietary factors” instead of “participants dietary factors”
Author Response
The authors investigated the relationship between serum zonulin levels and two pregnancy-specific metabolic diseases, intrahepatic cholestasis of pregnancy (ICP) and gestational diabetes mellitus (GDM), through a prospective case-control study. A total of 342 participants were enrolled in four groups: 95 ICP, 110 GDM, 16 ICP and GDM, and 136 healthy control. Participants were age-matched, came from the same geographic area, and were screened for exclusion criteria like chronic diseases. Blood samples were obtained from all participants in the third trimester, and human zonulin concentrations in sera were measured using Sun-Red Bio Company ELISA kits. After delivery, birth weight of the newborns and other data were collected to compare parameters. Then, statistical analyses were performed to test for differences between the groups.
- Strengths:
- The study had a large sample size and collected information on a number of parameters that could affect patient outcomes.
- The authors provided extensive statistical data in tables containing group baseline characteristics, pairwise comparisons, and the significance of any differences.
- The study design was clearly explained. The authors explained the unique aspects of this study compared with others in the discussion section, like including a group for patients with both GDM and ICP.
- The authors acknowledge some limitations of the study, like not controlling for dietary factors or measuring fecal zonulin levels.
Answers: Thank you for your valuable comments about the manuscript.
Major
- The authors mention some limitations of the study but do not acknowledge the potentially confounding statistically significant difference in BMI between the groups, a factor that could affect “leaky gut” and zonulin levels regardless of ICP or GDM diagnosis. The group with the highest zonulin levels also has the highest BMIs.
Answers: Thank you for your opinion, which adds an important dimension to our article. As you mentioned, it was detected in our results that the highest BMI level belonged to pregnant women having both GDM and ICP, than, GDM. This result is parallel to the current literature which indicates the obesity as a risk factor for GDM [Yang HY, Shariff ZM, Yosuf BNM, Rejali Z, Tee YYS, Bindels J, et al. Independent and combined effects of age, body mass index and gestational weight gain on the risk of gestational diabetes mellitus. Sci Rep 2020;10:8486.]. It was inferred from the results that obesity might trigger main metabolic diseases of pregnancy via inducing different pathophysiological mechanisms. This implication suggest the importance of weight control in pregnancy. Also, because BMI is not an adjustable and changable parameter, it constitutes a confounding factor. Due to rarity of pregnant women having ICP and GDM and ICP, the study population had to consist of participants with heterogen BMI levels. This is a limitation of the study.
- While there are many statistically significant results in the provided tables, the authors only really discuss the zonulin levels.It would be interesting to see their analysis of the other characteristics that differed between the groups.
Answers: Thank you for your valuable suggestion. We explained and discussed other significant results of the study in discussion section and marked-up with yellow colour.
- The discussion section was largely a summarization of other studies on this topic. While it may be relevant that the results were consistent with other studies, I would have expected a deeper analysis of the authors’ own results and their meaning in the context of ICP and GDM.
Answers: Thank you for your valuable suggestion. We explained and analyzed the results more detailed as you commented.
- The chosen ELISA Kit to measure Zonulin should be explained since measuring blood Zonulin is very difficult and critical too. Several Publications report on the disadvantages of this ELISA Kits.
Answers: Thank you for your valuable opinion that gives you the opportunity to explain this point in the article. We give detailed information about the ELISA kit that we used in material and methods section. Also it is easy to access and we think many health care services can use this kit more widely. Also we added some articles that used this ELISA kit in the literature:
- Demir E, Ozkan H, Seckin KD, Sahtiyancı B, Demir B. Plasma Zonulin Levels as a Non-Invasive Biomarker of Intestinal Permeability in Women with Gestational Diabetes Mellitus. Biomolecules 2019;24(9):1–8.
- Deniz CD, Ozler S, Sayın FK. Association of adverse outcomes of intrahepatic cholestasis of pregnancy with zonulin levels. J Obstet Gynaecol (Lahore) 2020;24:1–6.
- Mokkala K, Tertti K, Rönnemaa T, Vahlberg T, Laitinen K. Evaluation of serum zonulin for use as an early predictor for gestational diabetes. Nutr Diabetes 2017;7(3):2016–2018.
- Mokkala K, Pellonperä O, Röytiö H, Pussinen P, Rönnemaa T, Laitinen K. Increased intestinal permeability, measured by serum zonulin, is associated with metabolic risk markers in overweight pregnant women. Metabolism 2017;69(2017):43–50.
- Zak-Golab A, Kocelak P, Aptekorz M, Zeintara M, Juszczyk L, Matirosian G, et al. Gut microbiota, microinflammation, metabolic profile, and zonulin concentration in obese and normal weight subjects. International Journal of Endocrinology 2013;2013:674106.
Minor
- Line 3: “Coexistence”
- Line 8, 20: email font size larger than the rest
- Line 24: “intrahepatic” (lowercase i)
- Line 44: “liver disease, characterized by…”
- Line 47: I’m not sure what the authors mean by “contraception assumption”. Do they mean conception?
- Line 48: “ICP” instead of “The ICP”
- Line 50: “such as itching, nausea, loss of appetite, or pain in the…”
- Line 54: “Gestational diabetes mellitus” instead of “The gestational diabetes mellitus”
- Lines 60-61: The sentence feels incomplete: “In fact, previous studies have demonstrated the coexistence of GDM and ICP”. Even this study looks at people who have both GDM and ICP, proving that both can exist in the same person. Was there something more to this? Like a connection to the previous sentence about bile acids?
- Lines 75, 102, 230, 235, 236, 241, 247, 285: Replace “pregnants” with “participants”, “patients”, or “pregnant women”.
- Line 91: “including types 1” (missing a space)
- Line 94: “1) pruritus” (missing a space)
- Lines 104, 110: Replace “applied” with “given” or “provided”.
- Line 109: “mg/dL,” instead of “mg/dL.”
- Line 127: “zonulin in the samples” instead of “zonulin the samples”
- Line 138: “in Group 3” instead of “in the group 3”
- Line 138-139: “follow-ups were performed” instead of “follow-up was performed”
- Line 144: Consider listing weight groups in ascending order, with the first group the lowest weight (<2,500g) and the third group the highest (>4,000g).
- Line 192/Table 2: Consider ordering the rows left to right from lowest numerical comparison to highest. (Group 1-Group 2, then Group 1-Group 3, then Group 1-Group 4, and so on)
- Line 198/Figure 2: Consider listing the groups from top to bottom in ascending order (1,2,3,4) for consistency with other tables.
- Line 207: “(p=0.404)” instead of “(0.404)”
- Line 235: “gestational weeks” instead of “gestational week”
- Line 259: “study results” instead of study’ results”
- Line 280: “participants’ dietary factors” instead of “participants dietary factors”
Answers: Thank you for yout valuable contribution. All suggested cahnages were made marked-up with yellow colour in the paper.
Reviewer 4 Report
- In Figure 1, Flowchart of the study I would indicate Group 1, Group 2, etc so that it is connected with Table 1.
- All the P values are raw I would highly urge to correct all the P values for multiple testing.
-
Figure 2 title and labels of the plot is missing.
- Table 2 indicates Bonferoni corrected p values how ever the values are not exact. I would highly urge to get exact p values.
Author Response
In Figure 1, Flowchart of the study I would indicate Group 1, Group 2, etc so that it is connected with Table 1.
Answers: Thank you for your valuable suggestion. We made these changes and added to the figure.
All the P values are raw I would highly urge to correct all the P values for multiple testing.
Thank you for your valuable opinion that gives you the opportunity to explain this point in the article. All necessary statistical analysis applied to data were explained in material and methods section. Exact p values gathered as 0,000 were stated as <0,001.
Figure 2 title and labels of the plot is missing.
Answers: We added and confirmed these points of figure.
Table 2 indicates Bonferoni corrected p values how ever the values are not exact. I would highly urge to get exact p values.
Answers: Thank you for your opinion. As we compared 4 groups, we had to give Bonferoni corrected p values.
Round 2
Reviewer 3 Report
Thank you for your answers. However a proper discussion on the limitations of serum Zonulin Assays would increase the value of the paper substantially.
Author Response
Answer: thanks for reviewer’ comments about our manuscript. We added explanation about serum zonulin assays on limitations section.